# Health and dental care expenditures in the United States from 1996 to 2016

Man Hung [1,2,3,4,5] *, Martin S. Lipsky[1,6], Ryan Moffat[1], Evelyn Lauren[7], Eric S. Hon[8], Jungweon Park[1], Gagandeep Gill [1], Julie Xu[2], Lourdes Peralta[2], Joseph Cheever[1], David Prince[1], Tanner Barton[2], Nicole Bayliss[1], Weston Boyack[1], Frank W. Licari[1]

1 Roseman University of Health Sciences College of Dental Medicine, South Jordan, UT, United States of America, 2 University of Utah School of Medicine, South Jordan, UT, United States of America, 3 University of Utah School of Business, South Jordan, UT, United States of America, 4 University of Utah College of Education, South Jordan, UT, United States of America, 5 Towson University Department of Occupational Therapy & Occupational Science, Towson, MD, United States of America, 6 Portland State University College of Urban & Public Affairs, Portland, OR, United States of America, 7 Boston University Department of Biostatistics, Boston, MA, United States of America, 8 University of Chicago Department of Economics, Chicago, IL, United States of America

* mhung@roseman.edu

## Abstract

### Introduction

As total health and dental care expenditures in the United States continue to rise, healthcare disparities for low to middle-income Americans creates an imperative to analyze existing expenditures. This study examined health and dental care expenditures in the United States from 1996 to 2016 and explored trends in spending across various population subgroups.

### Methods

Using data collected by the Medical Expenditure Panel Survey, this study examined health and dental care expenditures in the United States from 1996 to 2016. Trends in spending were displayed graphically and spending across subgroups examined. All expenditures were adjusted for inflation or deflation to the 2016 dollar.

### Results

Both total health and dental expenditures increased between 1996 and 2016 with total healthcare expenditures increasing from $838.33 billion in 1996 to $1.62 trillion in 2016, a 1.9-fold increase. Despite an overall increase, total expenditures slowed between 2004 and 2012 with the exception of the older adult population. Over the study period, expenditures increased across all groups with the greatest increases seen in older adult health and dental care. The per capita geriatric dental care expenditure increased 59% while the per capita geriatric healthcare expenditure increased 50% across the two decades. For the overall US population, the per capita dental care expenditure increased 27% while the per capita healthcare expenditure increased 60% over the two decades. All groups except the uninsured experienced increased dental care expenditure over the study period.

**Data Availability Statement:** The data underlying the results presented in the study are available from a public depository at https://www.meps.ahrq.gov/mepsweb/data_stats/download_data_files.jsp. The data files are the "Full-Year

Consolidated Data Files" from the years 1996 to 2016.

**Funding:** The authors received no specific funding for this work.

**Competing interests:** The authors have declared that no competing interests exist.

## Conclusions

Healthcare spending is not inherently bad since it brings benefits while exacting costs. Our findings indicate that while there were increases in both health and dental care expenditures from 1996 to 2016, these increases were non-uniform both across population subgroups and time. Further research to understand these trends in detail will be helpful to develop strategies to address health and dental care disparities and to maximize resource utilization.

## Introduction

Over the past two decades, United States health and dental care expenditures have continued to increase in growth despite efforts to decrease spending. [1] Previous studies suggest that factors such as increased deductibles and copayments, improved delivery system efficiency, expiration of expensive drug patents, and evidenced-based approaches to costly medical innovations may have also contributed to the slowdown of healthcare expenditure. [2–4] However, low and middle-income Americans disproportionally experience the slowest spending growth, skewing overall health spending towards higher-income Americans [3] and widening the gap in spending between the poor and the wealthy. [5] While the Affordable Care Act (ACA) sought to address healthcare spending disparities by expanding Medicaid, inequalities in health spending persist. [6]

A key component of the Medicaid expansion was a mandatory dental benefit for children. [7] This adoption significantly changed the pediatric dental care landscape by no longer restricting dental coverage only to low income children. [7] While the ACA did not make adult dental care mandatory, it did provide opportunities for adults to obtain dental benefits through the insurance exchanges. [7, 8] As a result, about 8.3 million adults gained dental care benefits beyond emergency care since the ACA became law in 2010. [7] However, provider reimbursement and benefit coverage affect the likelihood of utilizing dental services and poor reimbursement for dentists participating in public health programs may contribute to the steady decline in dental care utilization among adults. [9, 10] As more senior citizens and low-income adults shift away from dental coverage to Medicaid, the ratio of Medicaid to private dental insurance plans has been reduced dramatically across 36 states and may impact dental expenditure. [7, 11, 12]

In 2012, annual healthcare expenditure in the United States amounted to $1.3 trillion with dental care representing approximately $85 billion of the total costs. [12] Of the $85 billion spent on dental care, Medicaid and Medicare accounted for $5.5 billion while $76 billion was either paid out-of-pocket or by private dental insurances. [2] Although dental costs are a relatively small percentage of the total expenditures, it remains a sizeable expenditure which should not be ignored. In addition, spending for oral health is an important issue since oral health contributes significantly to overall health and poor oral health is associated with chronic illnesses such as diabetes and heart disease.

Understanding health and dental care expenditures can help direct future planning and guide reform healthcare programs accordingly. [13–15] From the public payer perspective the anticipation that Medicaid enrollment will double or even triple makes it critically important to understand expenditure trends. [7, 16] Though previous research examined overall health and dental care expenditures, these studies were limited because their research design focused on limited samples, settings and narrow time frames. [17, 18] This study adds to the literature by using national samples that are representative of the United States' population to describe

**Table 1. Demographic characteristics.**

| Variable | Description | N* | %* | Mean | SD | Median |
|---|---|---|---|---|---|---|
| AGE | Age (year) | 684,381 | | 34.5 | 22.4 | 33.0 |
| TTLP | Person's total income ($) | 690,298 | | 26,070 | 30,556 | 17,960 |
| INSCOV | Insurance coverage | | | | | |
| | Private | 392,680 | 56.9 | | | |
| | Public | 188,316 | 27.3 | | | |
| | Uninsured | 109,302 | 15.8 | | | |
| SEX | Sex | | | | | |
| | Male | 329689 | 47.8 | | | |
| | Female | 360609 | 52.2 | | | |
| MARRY | Marital status | | | | | |
| | Married | 257,494 | 37.3 | | | |
| | Widowed | 33,302 | 4.8 | | | |
| | Divorced | 53,923 | 7.8 | | | |
| | Separated | 13,404 | 1.9 | | | |
| | Never married | 155,357 | 22.5 | | | |
| | Under 16 –N/A | 176,283 | 25.6 | | | |

trends in health and dental care expenditures over the past two decades. It further augments the literature by exploring trends both before and after the global financial crisis of 2007 to 2012. In addition, we investigated the trends of expenditures across demographic groups sand insurance coverage.

**Table 2. Total healthcare expenditures in the United States.**

| Year | Total healthcare expenditures | Total healthcare expenditures (Adjusted to 2016 US dollars) | Total geriatric healthcare expenditures | Total geriatric healthcare expenditures (Adjusted to 2016 US dollars) |
|---|---|---|---|---|
| 1996 | $548.04 billion | $838.33 billion | $158.94 billion | $243.12 billion |
| 1997 | $553.24 billion | $827.30 billion | $169.98 billion | $254.19 billion |
| 1998 | $560.59 billion | $825.44 billion | $171.97 billion | $253.22 billion |
| 1999 | $596.07 billion | $858.71 billion | $178.83 billion | $257.63 billion |
| 2000 | $627.89 billion | $875.14 billion | $182.31 billion | $254.09 billion |
| 2001 | $726.35 billion | $984.92 billion | $218.05 billion | $295.67 billion |
| 2002 | $810.72 billion | $1.08 trillion | $248.17 billion | $331.09 billion |
| 2003 | $895.52 billion | $1.17 trillion | $274.89 billion | $358.57 billion |
| 2004 | $963.88 billion | $1.22 trillion | $298.48 billion | $379.23 billion |
| 2005 | $1.02 trillion | $1.26 trillion | $303.33 billion | $372.77 billion |
| 2006 | $1.03 trillion | $1.23 trillion | $304.97 billion | $363.07 billion |
| 2007 | $1.12 trillion | $1.30 trillion | $334.67 billion | $387.39 billion |
| 2008 | $1.14 trillion | $1.28 trillion | $342.68 billion | $382.00 billion |
| 2009 | $1.26 trillion | $1.40 trillion | $363.58 billion | $406.75 billion |
| 2010 | $1.26 trillion | $1.39 trillion | $386.27 billion | $425.15 billion |
| 2011 | $1.33 trillion | $1.41 trillion | $394.93 billion | $421.39 billion |
| 2012 | $1.35 trillion | $1.41 trillion | $395.54 billion | $413.48 billion |
| 2013 | $1.40 trillion | $1.44 trillion | $419.99 billion | $432.69 billion |
| 2014 | $1.50 trillion | $1.52 trillion | $478.17 billion | $484.78 billion |
| 2015 | $1.60 trillion | $1.62 trillion | $520.73 billion | $527.30 billion |
| 2016 | $1.62 trillion | $1.62 trillion | $566.75 billion | $566.75 billion |

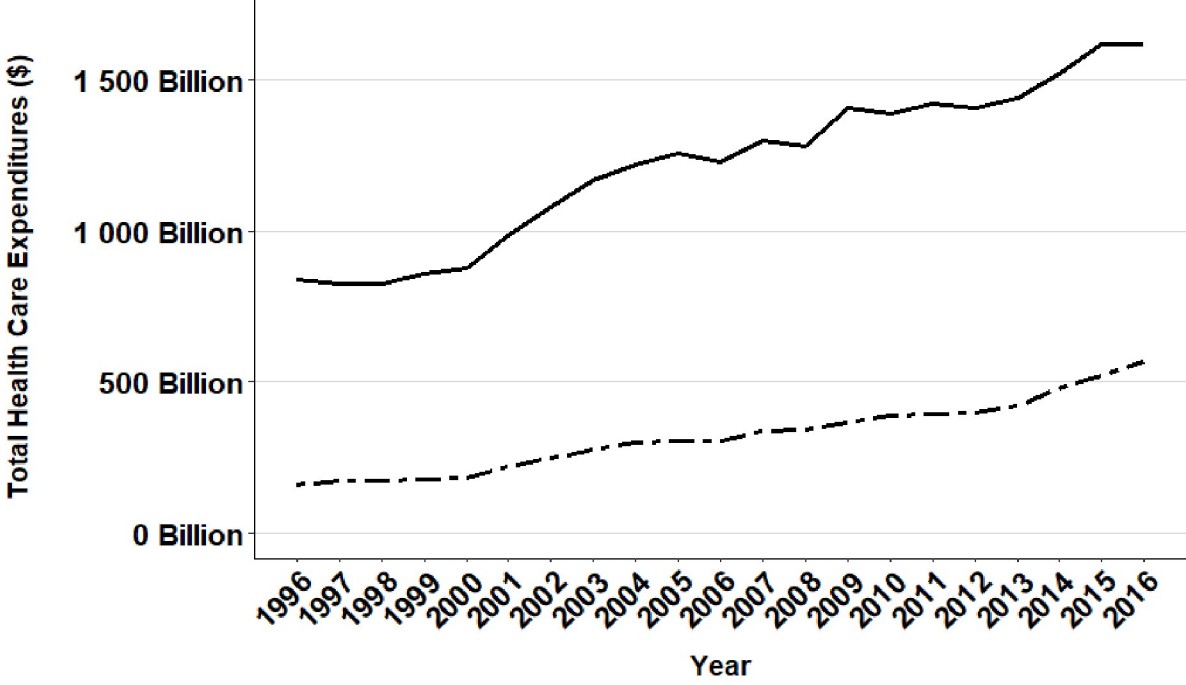

**Fig 1. Total health care expenditures in the United States (Entire population versus 65 years and older).**

## Methods

This study was descriptive in nature and examined health and dental care expenditures from 1996 to 2016 for the United States noninstitutionalized population. Longitudinal data were obtained from the 1996–2016 Medical Expenditure Panel Survey (MEPS) Household Component, a nationally representative study of the United States civilian, noninstitutionalized population collected annually and sponsored by the Agency for Healthcare Research and Quality. [11] MEPS provides information about a respondent's demographics, socioeconomic characteristics, health state and access to care and represents a complete data source on the health and dental expenditures by individuals and families in the nation. A more detailed description of MEPS is provided at https://www.meps.ahrq.gov/. Per federal regulations (45 CFR 46, category 4), this study is deemed exempt and does not require review from Institutional Review Board since the data were deidentified and publicly available.

Initial data processing began with merging all data from the 1996 to 2016 MEPS. Demographic characteristics were examined for all respondents from 1996 to 2016. Descriptive statistics of health and dental care expenditures were calculated and graphical representations displayed, stratifying by various groups such as age, gender, marital status, race, income and insurance coverage. Health and dental care expenditures included expenses such as inpatient and outpatient visits and drug expenditure. The per capita average expenditure as well as total expenditures were computed. For the total expenditures, the sum of all expenditures is reported. Total income was calculated based on the working age population from 15 to 64 years old. To compute the statistics for the geriatric population, we defined this group as adults aged 65 years and over.

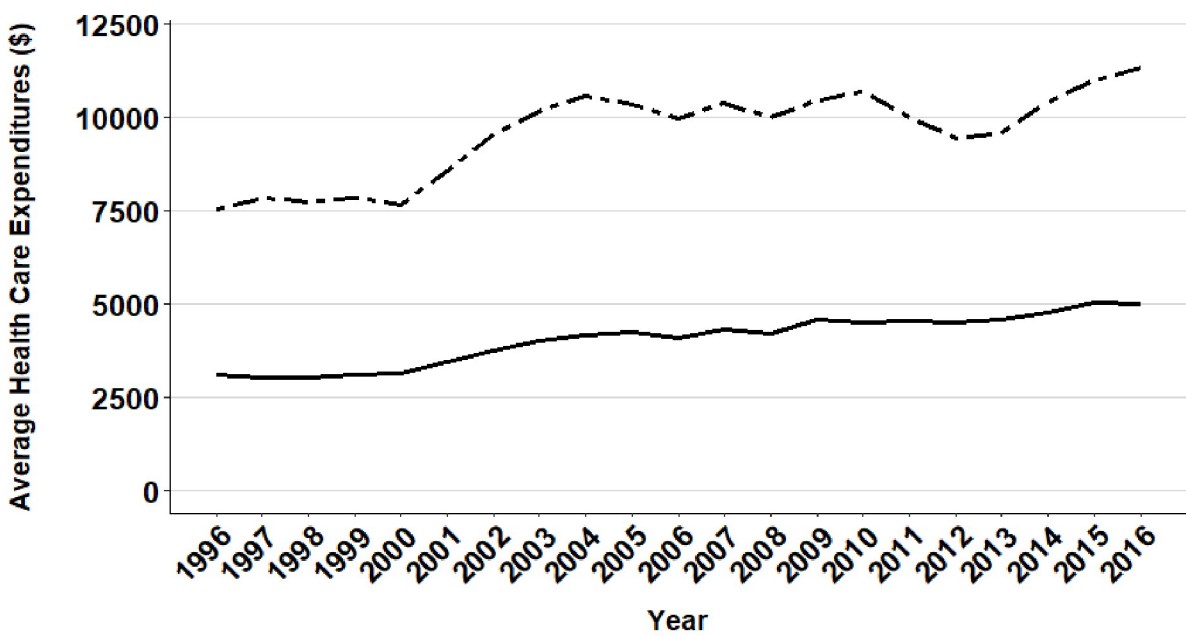

**Fig 2. Average health care expenditure per person in the United States (Entire population versus 65 years and older).**

In order to account for the effect of inflation or deflation over the study period, we adjusted all expenditures to the dollar values reflective of the year 2016 using inflation rates (or deflation rates when appropriate) published by the United States Bureau of Labor Statistics. MEPS uses a complex sample design including stratification, clustering and disproportionate sampling. The survey design was taken into consideration, and to enable the study findings to be representative of the United States population, we applied sampling weights to estimate expenditures in the population. Application of sampling weights is warranted because some population subgroups could have small or disproportionate sample sizes and applying sampling technique can provide sufficient sample size for statistical analyses leading to better precision. [19]

## Results

### Demographics

Table 1 summarizes the demographic and economic characteristics of the study population which consisted of 690,298 participants over the twenty-year period of 1996 to 2016. The mean age was 34.5 years old (standard deviation = 22.4). Approximately one-fourth of the study population was age 16 and younger, and 52% were female. The mean income was $26,070 (standard deviation = $30,556) and 57% of the population had private health insurance coverage.

### Healthcare expenditure

There was an increasing trend in total healthcare expenditures in the United States between 1996 and 2016 across all groups with a 1.9-fold increase from $838.33 billion in 1996 to $1.62 trillion in 2016 (Table 2, Fig 1). Over the same time period, healthcare expenditure for older

**Table 3. Average healthcare expenditure per person in the United States.**

| Year | Average healthcare expenditure | Average healthcare expenditure (Adjusted to 2016 US dollars) | Average geriatric healthcare expenditure | Average geriatric healthcare expenditure (Adjusted to 2016 US dollars) |
|---|---|---|---|---|
| 1996 | $2,038 | $3,118 | $4,922 | $7,530 |
| 1997 | $2,039 | $3,050 | $5,238 | $7,832 |
| 1998 | $2,049 | $3,018 | $5,253 | $7,734 |
| 1999 | $2,156 | $3,107 | $5,455 | $7,859 |
| 2000 | $2,255 | $3,143 | $5,483 | $7,643 |
| 2001 | $2,555 | $3,465 | $6,322 | $8,571 |
| 2002 | $2,813 | $3,753 | $7,158 | $9,551 |
| 2003 | $3,082 | $4,020 | $7,787 | $10,158 |
| 2004 | $3,284 | $4,172 | $8,333 | $10,587 |
| 2005 | $3,457 | $4,248 | $8,415 | $10,343 |
| 2006 | $3,452 | $4,110 | $8,368 | $9,965 |
| 2007 | $3,737 | $4,326 | $8,970 | $10,383 |
| 2008 | $3,773 | $4,206 | $8,975 | $10,007 |
| 2009 | $4,107 | $4,595 | $9,373 | $10,485 |
| 2010 | $4,094 | $4,507 | $9,702 | $10,678 |
| 2011 | $4,277 | $4,564 | $9,397 | $10,027 |
| 2012 | $4,309 | $4,504 | $9,029 | $9,438 |
| 2013 | $4,436 | $4,570 | $9,327 | $9,610 |
| 2014 | $4,708 | $4,774 | $10,269 | $10,412 |
| 2015 | $4,978 | $5,040 | $10,861 | $10,999 |
| 2016 | $5,006 | $5,006 | $11,358 | $11,358 |

adults increased by 2.3-fold from $243.12 billion in 1996 to $566.75 billion in 2016 (Table 2, Fig 1), making the increase in health expenditure approximately one-fourth greater for older adults compared to the general adult population.

Although between 1996 and 2016 total health expenditures demonstrated an overall increase, there was not a consistent annual upward trend. In fact, total healthcare expenditures experienced a steep increase from 2000 to 2004 (at a rate of $31.29 billion per year), and then slowed from 2004 to 2011 (to a rate of $6.02 billion per year), and resumed a steep upward trend again from 2012 to 2016 (to a rate of $38.56 billion per year) (Fig 3). Overall, the average healthcare expenditure per person in the United States increased by over 60% during the two decades, from $3,118 in 1996 to $5,006 in 2016 (Table 3, Fig 2). For the elderly population, the per capita increase in healthcare expenditure was over 50% spanning the two decades, from $7,530 in 1996 to $11,358 in 2016 (Table 3).

Fig 3 presents the annual average healthcare expenditure per person across subgroups in the United States. Those who were widowed had the highest per person healthcare spending throughout all twenty years, followed closely by the geriatric population. In recent years, divorced individuals rose more than single or married individuals in healthcare spending. The uninsured group had the lowest per person healthcare spending across the two decades. Those with public healthcare insurance spent substantially more than those with private health insurance and females spent substantially more than males. Asians demonstrated lower average healthcare spending compared to Caucasians and African Americans.

## Dental care expenditure

From 1996 to 2016, total dental care expenditures across all demographic groups increased by about 1.5-fold from $66.00 billion in 1996 to $101.26 billion in 2016 (Table 4). In contrast,

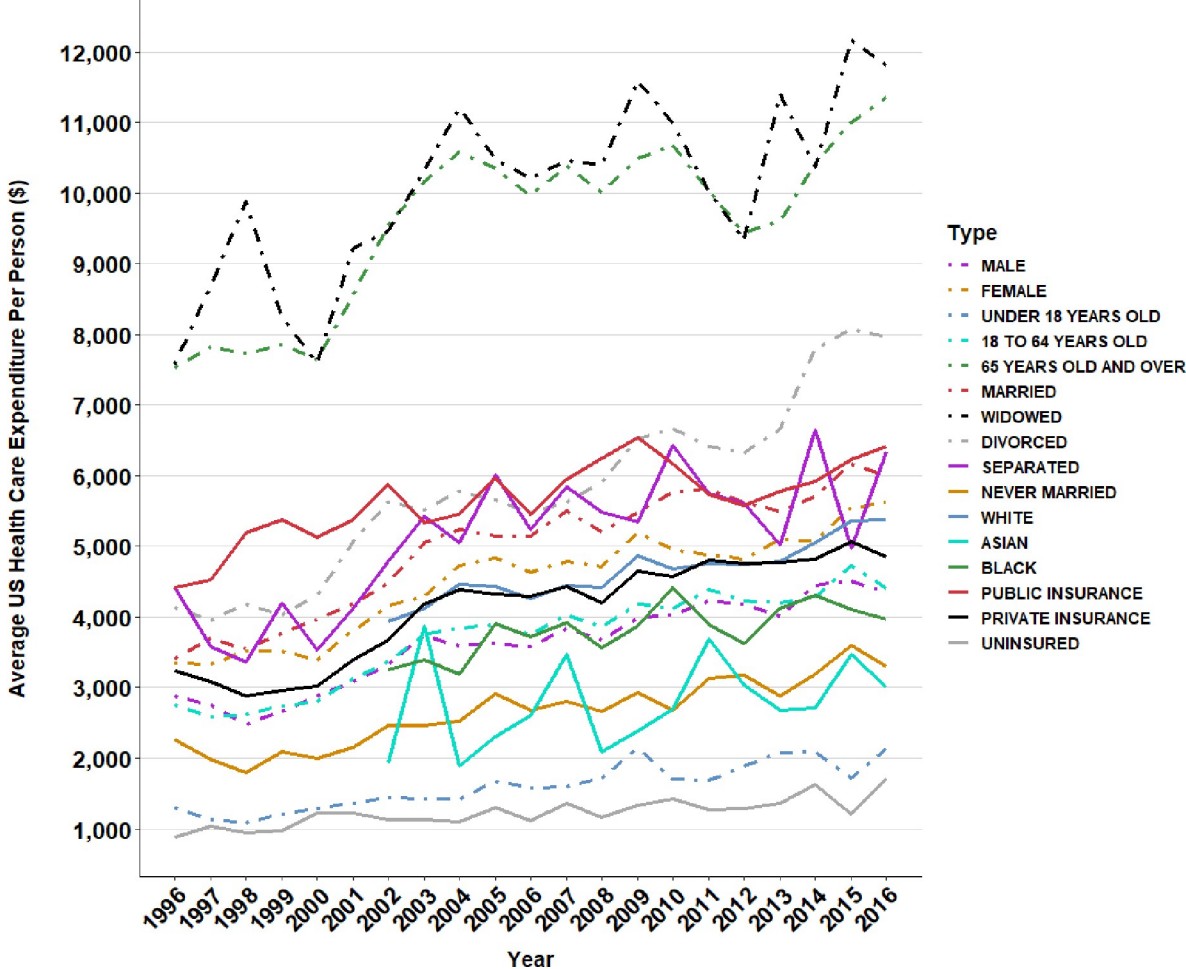

**Fig 3. Annual average health care expenditure per person across subgroups in the United States.**

over the same time period, total geriatric dental care expenditures increased 2.5-fold from $9.14 billion to $22.51 billion.

Although dental expenditure showed an increase over the study period, similar to overall healthcare expenditure, the increase was also non-linear and included periods of decline. Total dental expenditures decreased from $91.19 billion in 2004 to $89.06 billion in 2005 (Table 4, Fig 4). Interestingly, dental expenditure exhibited a marked drop from 2007 to 2012, where total dental expenditures decreased $6.3 billion from $94.96 billion to $88.66 billion, paralleling the global economic trend of the Great Recession. However, total geriatric dental expenditures did not show a similarly substantial drop over the 2007 to 2012 time span (Fig 4). Instead the data revealed three time periods each with relatively smaller decreases. Geriatric dental care expenditure decreased from $9.14 billion in 1996 to $8.46 billion in 1997, from $13.88 billion in 2005 to $13.29 billion in 2006, and from $15.20 billion in 2008 to $14.50 billion in 2009. Despite these three periods of small declines, total geriatric dental expenditures experienced an overall increase.

In 2015, dental expenditure displayed a dramatic $4 billion increase, the largest annual increase in cost and one that resulted in total dental expenditures rising to $97.12 billion. From 1996 to 2016, average dental expenditure per person in the United States paralleled

**Table 4. Total dental care expenditures in the United States.**

| Year | Total dental care expenditures | Total dental care expenditures (Adjusted to 2016 US dollars) | Total geriatric dental care expenditures | Total geriatric dental care expenditures (Adjusted to 2016 US dollars) |
|---|---|---|---|---|
| 1996 | $43.14 billion | $66.00 billion | $5.97 billion | $9.14 billion |
| 1997 | $44.50 billion | $66.54 billion | $5.66 billion | $8.46 billion |
| 1998 | $49.04 billion | $72.21 billion | $6.52 billion | $9.60 billion |
| 1999 | $52.20 billion | $75.20 billion | $7.01 billion | $10.10 billion |
| 2000 | $55.55 billion | $77.42 billion | $7.33 billion | $10.22 billion |
| 2001 | $59.49 billion | $80.67 billion | $7.64 billion | $10.36 billion |
| 2002 | $64.30 billion | $85.78 billion | $8.53 billion | $11.38 billion |
| 2003 | $66.97 billion | $87.35billion | $8.70 billion | $11.35 billion |
| 2004 | $71.77 billion | $91.19 billion | $9.85 billion | $12.52 billion |
| 2005 | $72.47 billion | $89.06 billion | $11.30 billion | $13.88 billion |
| 2006 | $76.32 billion | $90.86 billion | $11.16 billion | $13.29 billion |
| 2007 | $82.04 billion | $94.96 billion | $12.94 billion | $14.98 billion |
| 2008 | $84.08 billion | $93.73 billion | $13.64 billion | $15.20 billion |
| 2009 | $83.39 billion | $93.29 billion | $12.96 billion | $14.50 billion |
| 2010 | $83.11 billion | $91.48 billion | $14.11 billion | $15.53 billion |
| 2011 | $85.15 billion | $90.85 billion | $14.89 billion | $15.89 billion |
| 2012 | $84.81 billion | $88.66 billion | $16.92 billion | $17.69 billion |
| 2013 | $91.83 billion | $94.60 billion | $17.20 billion | $17.72 billion |
| 2014 | $91.87 billion | $93.14 billion | $19.61 billion | $19.88 billion |
| 2015 | $95.91 billion | $97.12 billion | $20.97 billion | $21.23 billion |
| 2016 | $101.26 billion | $101.26 billion | $22.51 billion | $22.51 billion |

trends in total dental care expenditures for the overall population. Average per person dental expenditure grew from $245.30 in 1997 to $310.69 in 2004 (Table 5, Fig 5), but started to decrease in 2008 and lowered to $292.52 in 2014, being comparable to where they stood in the early 2000's. Across the two decades, the average dental expenditure per person increased 27% from $245.44 to $313.37 for the overall population, and increased 59% from $283.29 to $451.02 for the elderly population.

Fig 6 shows the annual average dental care expenditure per person across subgroups in the United States. In most cases, the average dental care expenditure followed similar patterns with the exception of a few groups whose patterns differed from the average healthcare expenditure. The geriatric population experienced higher dental spending throughout the two decades, rising to the top in recent years. Those with private insurance also had very high dental expenditure and closely matched the geriatric population. Those with public health insurance spent substantially less in dental care than those with private health insurance. Caucasians experienced higher dental care expenditure compared to Asians and African Americans. Those who were widowed and those who were divorced experienced wider swings in spending with more dramatic increases and decreases in dental spending throughout the past two decades.

Fig 7 displays different sources of coverage for dental care expenditures. It is evident that the dental care required the most from out-of-pocket (e.g., self/family amount) for the past two decades. Private insurance coverage came very close to out-of-pocket dental expenses in the recent years.

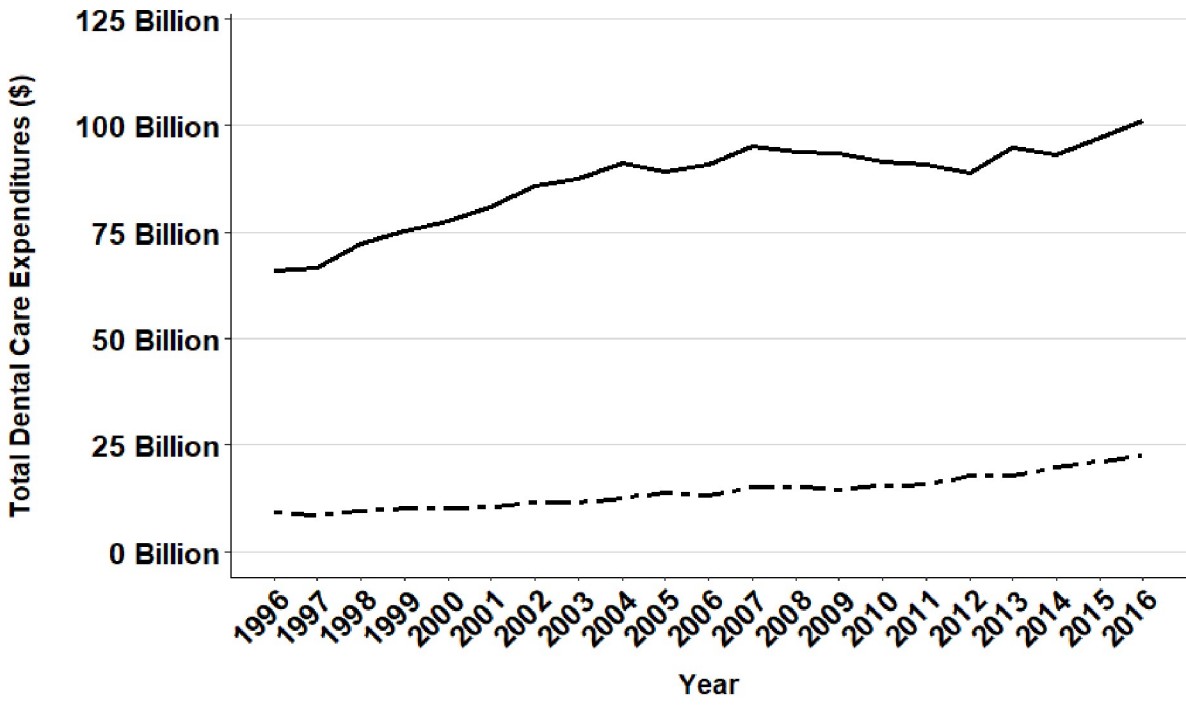

**Fig 4. Total dental care expenditures in the United States (Entire population versus 65 years and older).**

## Discussion

This study captures the most recent trends in health and dental care expenditures before and after the 2007–2009 financial crisis, and represents the most comprehensive longitudinal study examining health and dental care expenditures disaggregated by subgroups. Overall, both health and dental care expenditures per person have increased over the last two decades. Using a nationally representative database, we found that per person dental expenditure increased from $245 in 1999 to $313 in 2016 representing an average annual growth rate of 2.2%, though the growth rate for each year was not uniform. The decreases seen in 2007 to 2012 are most likely a consequence of the United States financial crisis, since annual spending started to increase again in the following years. However, average dental expenditure decreased over 10% from 2007 to 2012, while in contrast the average healthcare expenditure increased more than 4% over the same period suggesting that dental care may be more sensitive to an economic downturn than medical care. Understanding why this occurred will be important in the event of a future economic downturn. Another key finding was the inconsistent growth of dental expenditure across different social demographics. This study found the largest increases in spending in individuals 65 years and older, where dental expenditure per person nearly doubled with total healthcare expenditures nearly tripling.

Understanding which population groups drive health and dental care expenditures is important for addressing disparities in care and in balancing cost and benefit. The inconsistencies in spending across groups may reflect factors such as the healthcare environment or social barriers rather than major changes in medical or dental practice. For example, the influx of millions of Baby Boomers could create an increased demand for health and dental services that might affect the per capita cost of care. Since more older adults retain teeth due to advances in

**Table 5. Average dental care expenditure per person in the United States.**

| Year | Average dental care expenditure | Average dental care expenditure (Adjusted to 2016 US dollars) | Average geriatric dental care expenditure | Average geriatric dental care expenditure (Adjusted to 2016 US dollars) |
|------|------|------|------|------|
| 1996 | $160.45 | $245.44 | $185.15 | $283.29 |
| 1997 | $164.04 | $245.30 | $174.49 | $260.88 |
| 1998 | $179.30 | $264.01 | $199.33 | $293.48 |
| 1999 | $188.86 | $272.08 | $213.89 | $308.16 |
| 2000 | $199.53 | $278.10 | $220.69 | $307.60 |
| 2001 | $209.32 | $283.83 | $221.53 | $300.31 |
| 2002 | $223.14 | $297.69 | $246.15 | $328.45 |
| 2003 | $230.46 | $300.61 | $246.54 | $321.59 |
| 2004 | $244.53 | $310.69 | $275.17 | $349.60 |
| 2005 | $244.70 | $300.72 | $313.51 | $385.35 |
| 2006 | $255.03 | $303.62 | $306.48 | $364.95 |
| 2007 | $272.29 | $315.19 | $346.98 | $401.64 |
| 2008 | $276.25 | $307.95 | $357.30 | $398.39 |
| 2009 | $271.96 | $304.25 | $334.27 | $373.91 |
| 2010 | $269.35 | $296.46 | $354.51 | $390.16 |
| 2011 | $273.70 | $292.03 | $354.52 | $378.30 |
| 2012 | $270.56 | $282.83 | $386.26 | $403.80 |
| 2013 | $290.86 | $299.66 | $382.01 | $393.58 |
| 2014 | $288.53 | $292.52 | $421.13 | $426.99 |
| 2015 | $298.42 | $302.18 | $437.35 | $442.91 |
| 2016 | $313.37 | $313.37 | $451.02 | $451.02 |

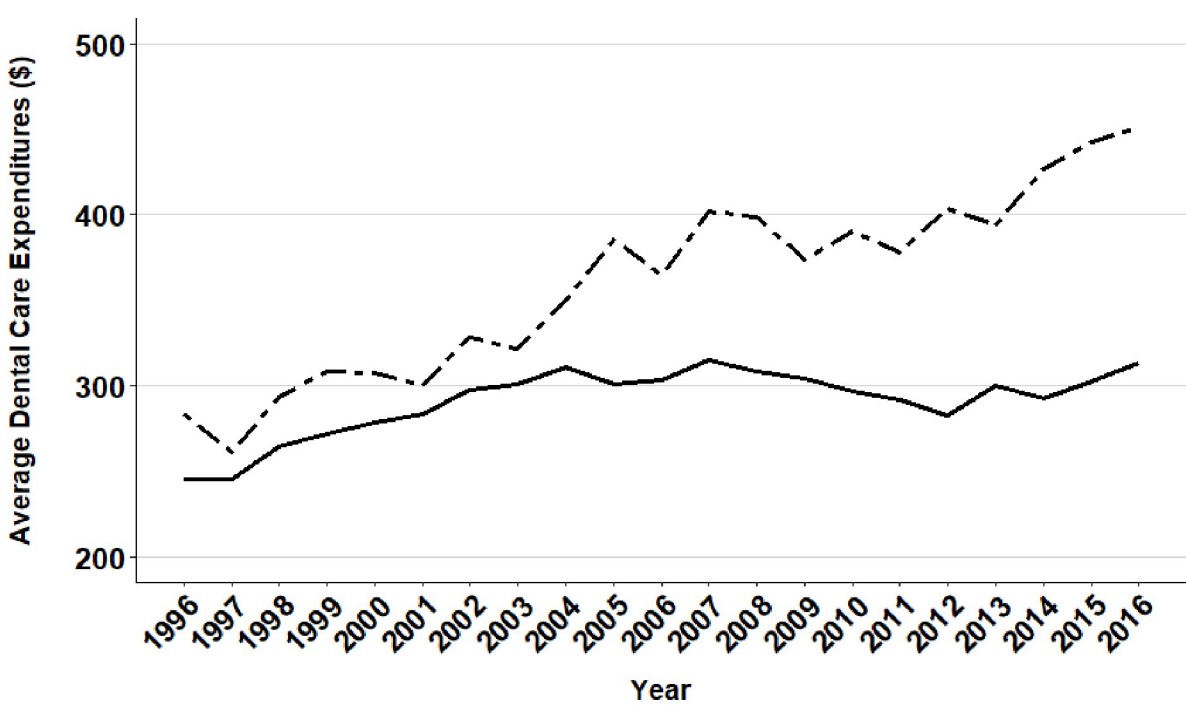

**Fig 5. Average dental care expenditure per person in the United States (Entire population versus 65 years and older).**

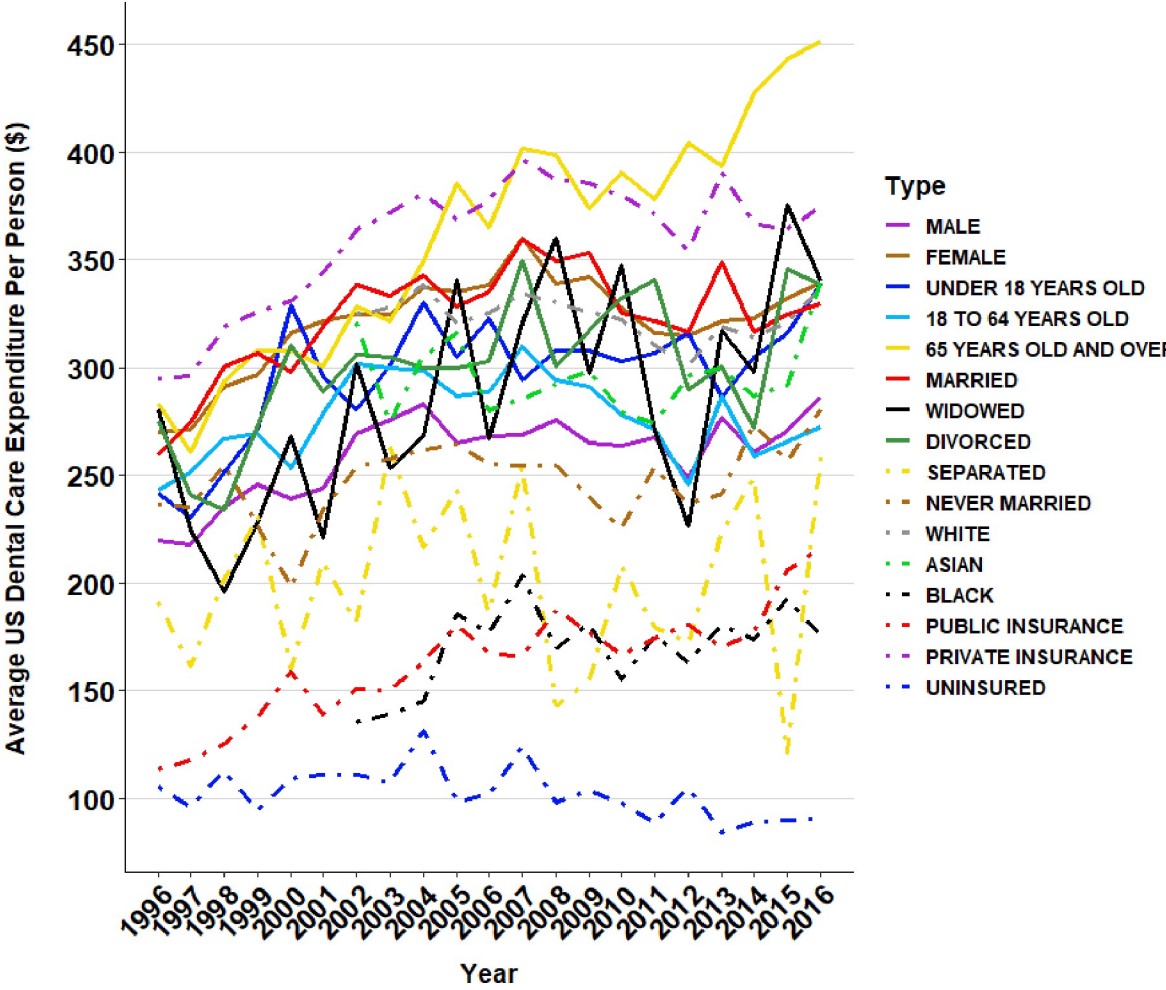

**Fig 6. Annual average dental care expenditure per person across subgroups in the United States.**

health and dental care, geriatric dental expenditure could increase. Moreover, enhanced application of dental implants that support prosthetic treatment have become more affordable over the last twenty years. In addition, more general dentists receive implant training, making it a more commonly used procedure and potentially contributing to the observed growth in dental expenditures.

Lower dental care spending for those with public health insurance and high out-of-pocket dental expenses raise questions about appropriately matching need to spending and whether Medicare and Medicaid system-level reform which includes sufficient dental coverage might be a cost-effective strategy to improve the nation's health. After all, oral health is the gateway to an individual's overall health and well-being and an investment in oral health could potentially reduce overall healthcare expenditure. [19] One example of investment benefiting both healthcare expenditure and outcomes is the reduction of acute cardiac events attributed to the wider use of use of medications for cardiovascular risk factors. Public healthcare financing policies that endorse Medicare and Medicaid dental care insurance coverage for citizens with unmet dental needs might yield a similar benefit. For example, seniors without adequate insurance might avoid preventive care or delay dental care, potentially shifting expenditures to costlier treatments. Future research targeted at dissecting the results reported by this study will be

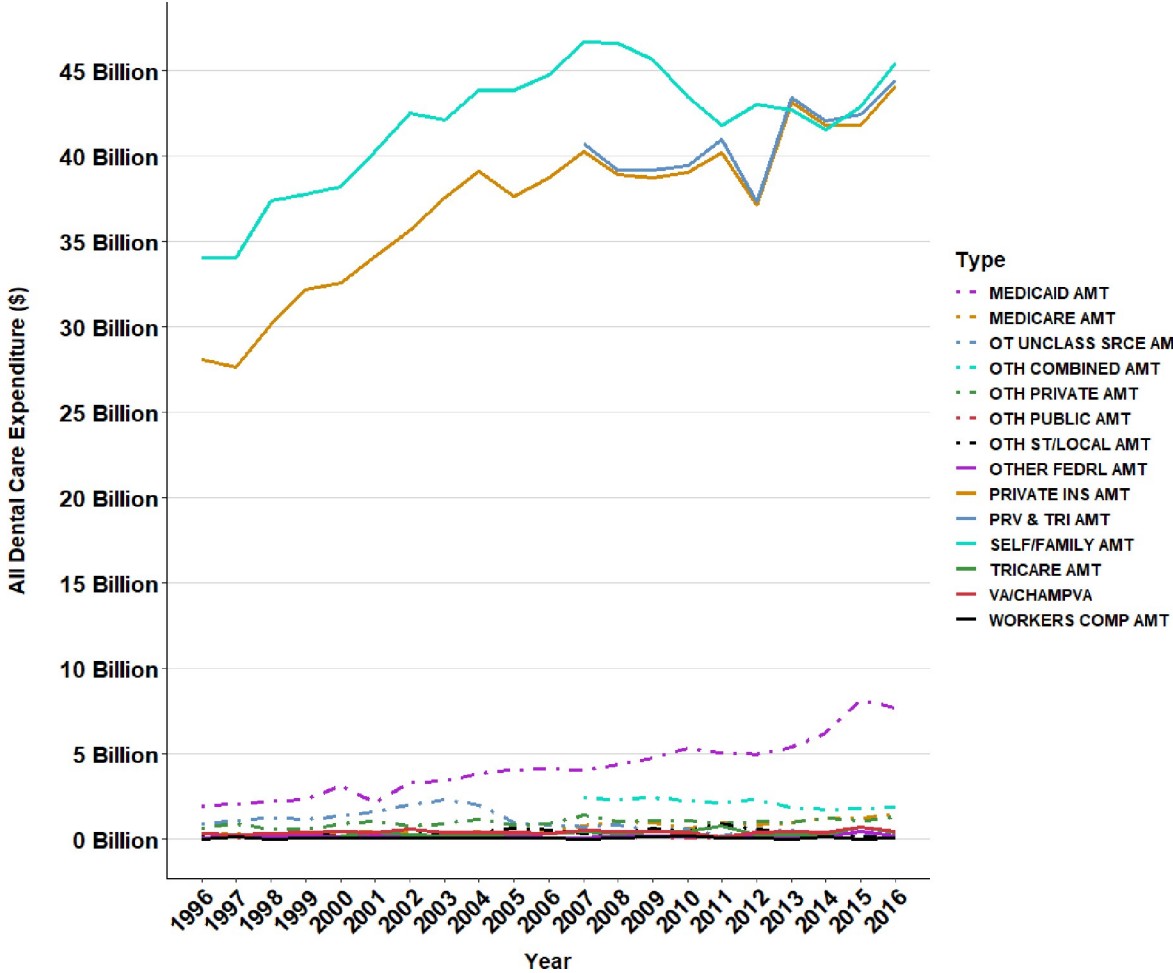

**Fig 7. Different sources of coverage for dental care expenditures.**

helpful in directing policy and strategies to maximize the health and dental services cost benefit ratio for individuals with unmet needs.

The trends identified by this study raise questions about the drivers of health expenditure that are important for assessing administrative costs and for setting targets to avoid wasteful spending. This study also adds to existing knowledge about disparities in expenditures across population groups, by characterizing where gaps keep widening [20–22] and providing useful benchmarks to evaluate policy interventions. Traditionally, older adults face challenges accessing dental care [23, 24] despite programs aimed at improving dental care plans for older patients. [23–25] While older adults receiving more dental care is certainly positive, expanding healthcare costs make measuring cost/benefit ratios important. As the uninsured or underinsured populations experience lower expenditures and unmet dental care, they also provide a standard to gauge the impact of interventions such as publicly funded portable and mobile dentistry clinics introduced by the 2017 Action for Dental Health Act. Funding these clinics could theoretically decrease healthcare expenditure by reducing the non-emergency burden on emergency departments and restore productivity lost to dental disease affecting underserved adult and pediatric populations. [26] Again, future research assessing the impact of these interventions on expenditures will be important for directing funds and policies.

## Limitations

Like all studies, this study does have limitations. The study used data from the civilian, noninstitutionalized population collected by MEPS, thus the interpretation should not be extended to the institutionalized population. Since MEPS collected data from noninstitutionalized population, our study did not include groups such as nursing home residents who are often a high cost and older population. Second, MEPS expenditures are lower than the National Health Expenditure Accounts database suggesting that the dependence on household reporting might result in an underreporting of individual medical events. [27] Another issue is that household reporting is potentially subject to misreporting because of a lack of technical knowledge. [27] However, a strength of MEPS is that it represents the most complete medical expenditure database in the United States and the results of this study should be generalizable to the United States population. Another limitation is that the expenditure data reported in this study represent accounting cost but not necessarily including economic cost such as sunk cost. [28] Other potential limitations include the differences in service charges by clinics or practices and sampling variations.

## Conclusions

Healthcare spending is affected by many factors including the overall economy, regulations and legislation. For this reason, understanding trends and disparities is important to the society. Our findings indicate that while there are increases in both health and dental care expenditures, these increases are non-uniform both across population subgroups and time. One apparent trend is that dental care is more influenced by changes in the economy than medical expenditures. For the elderly population, the per capita dental care expenditures are growing much faster than the per capita medical expenditures across the two decades. Further research to understand these trends in detail will be helpful to develop strategies to address health and dental care disparities and to maximize resource utilization.

## Acknowledgments

The authors sincerely thank Bianca Ruiz-Negrón from the University of Utah for her assistance in manuscript preparation and submission. The authors also thank Roseman University College of Dental Medicine Clinical Outcomes Research and Education for support of this study.

## Author Contributions

**Conceptualization:** Man Hung, Martin S. Lipsky, Ryan Moffat, Eric S. Hon, Jungweon Park, Gagandeep Gill, David Prince, Frank W. Licari.

**Data curation:** Man Hung, Evelyn Lauren.

**Formal analysis:** Man Hung, Evelyn Lauren, Eric S. Hon.

**Funding acquisition:** Man Hung, Frank W. Licari.

**Investigation:** Man Hung, Martin S. Lipsky, Ryan Moffat, Eric S. Hon, Jungweon Park, Gagandeep Gill, Julie Xu, Lourdes Peralta, Joseph Cheever, Tanner Barton, Nicole Bayliss, Weston Boyack, Frank W. Licari.

**Methodology:** Man Hung, Evelyn Lauren, Eric S. Hon, Joseph Cheever, Frank W. Licari.

**Project administration:** Man Hung.

**Resources:** Man Hung, Eric S. Hon, Frank W. Licari.

**Software:** Man Hung.

**Supervision:** Man Hung, Frank W. Licari.

**Validation:** Man Hung, Martin S. Lipsky, Ryan Moffat, Eric S. Hon, Julie Xu, Lourdes Peralta, Joseph Cheever, Tanner Barton, Nicole Bayliss, Weston Boyack, Frank W. Licari.

**Visualization:** Man Hung.

**Writing – original draft:** Man Hung, Ryan Moffat, Evelyn Lauren, Eric S. Hon, Jungweon Park, Gagandeep Gill, Julie Xu, David Prince, Tanner Barton.

**Writing – review & editing:** Man Hung, Martin S. Lipsky, Ryan Moffat, Eric S. Hon, Jungweon Park, Julie Xu, Lourdes Peralta, Joseph Cheever, David Prince, Nicole Bayliss, Weston Boyack, Frank W. Licari.

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
