## [Decision Letter · Decision Letter 0]

3 Jan 2020

PONE-D-19-27729

Health and dental care expenditures in the United States from 1996 to 2016

PLOS ONE

Dear Dr. Hung,

Thank you for submitting your manuscript to PLOS ONE. After careful consideration, we feel that it has merit but does not fully meet PLOS ONE’s publication criteria as it currently stands. Therefore, we invite you to submit a revised version of the manuscript that addresses the points raised during the review process.

We would appreciate receiving your revised manuscript by Feb 17 2020 11:59PM. To enhance the reproducibility of your results, we recommend that if applicable you deposit your laboratory protocols in protocols.io, where a protocol can be assigned its own identifier (DOI) such that it can be cited independently in the future. For instructions see: http://journals.plos.org/plosone/s/submission-guidelines#loc-laboratory-protocols

We look forward to receiving your revised manuscript.

Kind regards,

Frédéric Denis, Ph.D.

Academic Editor

PLOS ONE

Journal Requirements:

2. Please note that according to our submission guidelines (http://journals.plos.org/plosone/s/submission-guidelines), outmoded terms and potentially stigmatizing labels should be changed to more current, acceptable terminology. For example: “Caucasian” should be changed to “white” or “of [Western] European descent” (as appropriate).

Reviewers' comments:

Reviewer's Responses to Questions

**Comments to the Author**

1. Is the manuscript technically sound, and do the data support the conclusions?

Reviewer #1: Yes

Reviewer #2: No

2. Has the statistical analysis been performed appropriately and rigorously? 

Reviewer #1: Yes

Reviewer #2: No

3. Have the authors made all data underlying the findings in their manuscript fully available?

Reviewer #1: No

Reviewer #2: Yes

4. Is the manuscript presented in an intelligible fashion and written in standard English?

Reviewer #1: Yes

Reviewer #2: Yes

5. Review Comments to the Author

Reviewer #1: A well written manuscript that highlights the increase in dental expenditure among US geriatric population. As the authors describe,it is important to gain insights on the type of dental expenses this population is incurring. I feel Figure 6 display could be improved by selecting distinct colors for the dotted lines. It is hard to distinguish between lines that have similar colors.

Reviewer #2: This paper examines the trend in medical and dental care spending in the United States for the period from 1996-2016. The paper uses MEPS data. Among the findings are significant spending increases for the older population, and a slowdown in dental spending during the Great Recession. Overall, the paper is on an interesting topic and shows significant promise. However, there is a very important limitation.

The key limitation is that the MEPS data need to be adjusted before they can be used in this fashion. As the authors note (reference 27), the MEPS data do not add to national totals. There are several reasons for this: (1) the institutionalized sample is largely omitted from MEPS: (2) high spenders are underrepresented in MEPS; and (3) not all instances of medical care utilization are reported to MEPS. These three issues mean that total spending in MEPS is significantly below that in the national health accounts. Further, the trends in total spending in MEPS differ from those in national surveys. A methodology has been developed to address the deficiencies in MEPS spending (Rosen et al., https://www.nber.org/papers/w23290). The authors should implement an adjustment like this before undertaking the trend analyses.

In considering the results, I prefer to consider the per capita measures instead of the total measures. The elderly population has increased in number greatly over time. Thus, spending growth will necessarily be more rapid for that group. At minimum, the authors should indicate how much of the relative growth in spending for the elderly is due to demographic changes compared to differential growth in spending per person.

6. PLOS authors have the option to publish the peer review history of their article (what does this mean?). If published, this will include your full peer review and any attached files.

Reviewer #1: Yes: Thankam Paul Thyvalikakath, DMD, PhD

Reviewer #2: No

---

## [Author Response · Author response to Decision Letter 0]

25 Feb 2020

Reviewer #1: A well written manuscript that highlights the increase in dental expenditure among US geriatric population. As the authors describe, it is important to gain insights on the type of dental expenses this population is incurring. I feel Figure 6 display could be improved by selecting distinct colors for the dotted lines. It is hard to distinguish between lines that have similar colors.

We appreciate reviewer #1’s time to review our manuscript. We also believe that this study can contribute valuable insights to the public. Per reviewer #1’s suggestion, we improved Figure 6 by using distinct colors to make the dotted lines stand out and have darkened all of the solid and dotted lines and labels to enhance the figure’s quality.

Reviewer #2: This paper examines the trend in medical and dental care spending in the United States for the period from 1996-2016. The paper uses MEPS data. Among the findings are significant spending increases for the older population, and a slowdown in dental spending during the Great Recession. Overall, the paper is on an interesting topic and shows significant promise. However, there is a very important limitation.

The key limitation is that the MEPS data need to be adjusted before they can be used in this fashion. As the authors note (reference 27), the MEPS data do not add to national totals. There are several reasons for this: (1) the institutionalized sample is largely omitted from MEPS: (2) high spenders are underrepresented in MEPS; and (3) not all instances of medical care utilization are reported to MEPS. These three issues mean that total spending in MEPS is significantly below that in the national health accounts. Further, the trends in total spending in MEPS differ from those in national surveys. A methodology has been developed to address the deficiencies in MEPS spending (Rosen et al., https://www.nber.org/papers/w23290). The authors should implement an adjustment like this before undertaking the trend analyses.

In considering the results, I prefer to consider the per capita measures instead of the total measures. The elderly population has increased in number greatly over time. Thus, spending growth will necessarily be more rapid for that group. At minimum, the authors should indicate how much of the relative growth in spending for the elderly is due to demographic changes compared to differential growth in spending per person.

We thank reviewer #2 for the valuable feedback. We recognize the concerns raised by the reviewer and noted these in the limitations section of the paper. (See below).

“The study used data from the civilian, noninstitutionalized population collected by MEPS, thus the interpretation should not be extended to the institutionalized population. Since MEPS collected data from noninstitutionalized population, our study did not include groups such as nursing home residents who are often a high cost and older population. Second, MEPS expenditures are lower than the National Health Expenditure Accounts database suggesting that the dependence on household reporting might result in an underreporting of individual medical events.(27) Another issue is that household reporting is potentially subject to misreporting because of a lack of technical knowledge.(27) However, a strength of MEPS is that it represents the most complete medical expenditure database in the United States and the results of this study should be generalizable to the United States population. Another limitation is that the expenditure data reported in this study represent accounting cost but not necessarily include economic cost such as sunk cost.(28) Other potential limitations include the differences in service charges by clinics or practices and sampling variation.”

However, despite these limitations we believe using MEPS data without an adjustment remains justifiable. Unlike some databases, MEPS uniquely captures personal level data for expenditures, insurance coverage and demographics for the US non-institutionalized population as a whole. AQHR considers MEPS as the most complete database source on healthcare and expense in the US. (1) The continuing design methods used by MEPS and the number of years of available data also makes it a valuable tool for assessing trends in expenditures (2), critical to this paper since one aim was to examine trends over time. Also, for oral health, MEPS provides accurate estimates for dental utilization. Mancek MD and Manski RJ et. al (3) compared three nationally represented databases and recommended that for overall accurate estimates MEPS should be used. 

Yet because of limitations in using MEPS reviewer #2 recommended a methodology to adjust for deficiencies in the MEPS database. (4) While we were aware of the papers publishing methods to adjust the MEPS database, we appreciate the reviewer bringing the Rosen paper to our attention. It is both an interesting and thoughtful approach. However, our concern is the suggested methodology relies heavily on several assumptions that we believe would introduce uncertainty and ambiguity to our analysis and potentially raises new questions. Specifically, in the discussion section the Rosen paper cited limitations to their proposed methodology and noted that their adjustment requires multiple assumptions all of which are subject to error. Other investigators attempting similar reconciliations (5, 6, and 7) yielded different reconciliation estimates, highlighting the difficulty, complexity and uncertainty of these adjustments. So, while we understand the rationale for the reviewer’s recommendation, we believe an adjustment as proposed in the Rosen paper adds an additional layer of complexity and adds more limitations that might detract from our paper. 

Furthermore, integrating additional databases potentially complicates the analysis of trends over the time period (1996 to 2016) in our paper. For example, for some of the databases (e.g., MCBS, the NNHS, the SISFCF and demographics of the US Army) that Rosen suggested for use in adjustment, the MCBS data prior to 2013 are not publicly available for free, only the 1997, 1999 and 2004 NNHS data are available, only the 1997 and 2004 SISFCF data are available, and the Office of the Army does not have the 1996 and 1997 demographic data publicly available. As one can see, it would be extremely complicated to incorporate these additional databases which have an enormous amount of missing information to adjust for MEPS. A lot of unverifiable assumptions will have to be made of, which can introduce a great level of uncertainty for the adjusted results.

However, the concerns expressed by the reviewer do raise questions that we plan to address in a future paper comparing trends in the populations not covered by MEPS to the trends identified in this paper. We believe that adjusting the data, as suggested by the reviewer for this future study merits a separate paper devoted to this topic.

Finally, we also appreciate the suggestion that results for older adults be reported per capita. We have added these results per reviewer #2’s suggestion throughout the paper using track changes.

References

1. Bernard D, Cowan C, Selden T, Cai L, Catlin A, Heffler S. Reconciling medical expenditure estimates from the MEPS and NHEA, 2007. Medicare & Medicaid Research Review. 2012;2(4).

2. Chu MC. STATISTICAL BRIEF# 420: The Uninsured in America, 1996-2012: Estimates for the US Civilian Noninstitutionalized Population under Age 65. Agency for Healthcare Research and Quality. https://www.ncbi.nlm.nih.gov/books/NBK493868/. Published 2013. Accessed 2/5, 2020.

3. Macek MD, Manski RJ, Vargas CM, Moeller JF. Comparing oral health care utilization estimates in the United States across three nationally representative surveys. Health Serv Res. 2002;37(2):499–521. doi:10.1111/1475-6773.034

4. Medical Expenditure Panel Survey. Agency for Healthcare Research and Quality. https://www.meps.ahrq.gov/mepsweb/. Accessed 01/22, 2020.

5. Rosen AB, Ghosh K, Pape ES, et al. Strengthening National Data to Better Measure What We Are Buying in Health Care: Reconciling National Health Expenditures with Detailed Survey Data. National Bureau of Economic Research Working Paper Series. 2017; No. 23290.

6. Sing M, Banthin JS, Selden TM, Cowan CA, Keehan SP. Reconciling medical expenditure estimates from the MEPS and NHEA, 2002. Health care financing review. 2006;28(1):25.

7. Zuvekas SH, Olin GL. Accuracy of Medicare Expenditures in the Medical Expenditure Panel Survey. INQUIRY: The Journal of Health Care Organization, Provision, and Financing. 2009;46(1):92-108.

---

## [Decision Letter · Decision Letter 1]

28 May 2020

Health and dental care expenditures in the United States from 1996 to 2016

PONE-D-19-27729R1

Dear Dr. Hung,

We are pleased to inform you that your manuscript has been judged scientifically suitable for publication and will be formally accepted for publication once it complies with all outstanding technical requirements.

With kind regards,

Frédéric Denis, Ph.D.

Academic Editor

PLOS ONE

Additional Editor Comments (optional):

The major concern of this paper is the use of  the Medical Expenditure Panel Survey (MEPS) data base which is criticized. The author highlighted the limitations of the MEPS in the revised version and justified these methodological choices. The reviewer suggested the Rosen paper but this paper had several limitations. As for me, the MEPS data base represents the most complete medical expenditure database in the United States and The continuing design methods used by MEPS and the number of years of available data also makes it a valuable tool for assessing trends in expenditures, critical to this paper since one aim was to examine trends over time.

All methodological approaches have pros and cons and this paper can be accepted.

Reviewers' comments:

Reviewer's Responses to Questions

**Comments to the Author**

1. If the authors have adequately addressed your comments raised in a previous round of review and you feel that this manuscript is now acceptable for publication, you may indicate that here to bypass the “Comments to the Author” section, enter your conflict of interest statement in the “Confidential to Editor” section, and submit your "Accept" recommendation.

Reviewer #1: All comments have been addressed

Reviewer #2: (No Response)

2. Is the manuscript technically sound, and do the data support the conclusions?

Reviewer #1: Yes

Reviewer #2: Partly

3. Has the statistical analysis been performed appropriately and rigorously? 

Reviewer #1: I Don't Know

Reviewer #2: Yes

4. Have the authors made all data underlying the findings in their manuscript fully available?

Reviewer #1: Yes

Reviewer #2: Yes

5. Is the manuscript presented in an intelligible fashion and written in standard English?

Reviewer #1: Yes

Reviewer #2: Yes

6. Review Comments to the Author

Reviewer #1: (No Response)

Reviewer #2: I raised an issue in my earlier review that the data were not accurate because the MEPS is a selected sample. The authors gave two responses: (1) it would be difficult to fix the issues; and (2) there is error in any adjustment that might be made. The second of these is not a convincing critique. Of course there are errors in adjustments. But how does that justify using data that we know to be incorrect (and which, by the way, are also estimated and adjusted)?

The only real argument is the first one. For many possible adjustments, the correction would likely be small and could be omitted. But for people in skilled nursing facilities, the adjustment could be material. If I were looking for a paper to cite on trends in spending in the elderly population, I would not cite one that did not make an adjustment to MEPS for the institutionalized population.

7. PLOS authors have the option to publish the peer review history of their article (what does this mean?). If published, this will include your full peer review and any attached files.

Reviewer #1: Yes: Thankam Thyvalikakath

Reviewer #2: No

---

## [Editor Report · Acceptance letter]

3 Jun 2020

PONE-D-19-27729R1 

Health and dental care expenditures in the United States from 1996 to 2016 

Dear Dr. Hung:

I'm pleased to inform you that your manuscript has been deemed suitable for publication in PLOS ONE. Congratulations! Your manuscript is now with our production department. 

Kind regards, 

on behalf of

Dr. Frédéric Denis 

Academic Editor

PLOS ONE